# Partial Fluxes of Phosphoric Acid Anions through Anion-Exchange Membranes in the Course of NaH_2_PO_4_ Solution Electrodialysis

**DOI:** 10.3390/ijms20143593

**Published:** 2019-07-23

**Authors:** Olesya Rybalkina, Kseniya Tsygurina, Ekaterina Melnikova, Semyon Mareev, Ilya Moroz, Victor Nikonenko, Natalia Pismenskaya

**Affiliations:** Kuban State University, 149 Stavropolskaya st., 350040 Krasnodar, Russia

**Keywords:** ion-exchange membrane, Fujifilm, Neosepta, phosphate transport, limiting current density, voltammetry

## Abstract

Electrodialysis (ED) with ion-exchange membranes is a promising method for the extraction of phosphates from municipal and other wastewater in order to obtain cheap mineral fertilizers. Phosphorus is transported through an anion-exchange membrane (AEM) by anions of phosphoric acid. However, which phosphoric acid anions carry the phosphorus in the membrane and the boundary solution, that is, the mechanism of phosphorus transport, is not yet clear. Some authors report an unexpectedly low current efficiency of this process and high energy consumption. In this paper, we report the partial currents of H_2_PO_4_^−^, HPO_4_^2−^, and PO_4_^3−^ through Neosepta AMX and Fujifilm AEM Type X membranes, as well as the partial currents of H_2_PO_4_^−^ and H^+^ ions through a depleted diffusion layer of a 0.02 M NaH_2_PO_4_ feed solution measured as functions of the applied potential difference across the membrane under study. It was shown that the fraction of the current transported by anions through AEMs depend on the total current density/potential difference. This was due to the fact that the pH of the internal solution in the membrane increases with the growing current due to the increasing concentration polarization (a lower electrolyte concentration at the membrane surface leads to higher pH shift in the membrane). The HPO_4_^2−^ ions contributed to the charge transfer even when a low current passed through the membrane; with an increasing current, the contribution of the HPO_4_^2−^ ions grew, and when the current was about 2.5 *i*_lim_^Lev^ (*i*_lim_^Lev^ was the theoretical limiting current density), the PO_4_^3−^ ions started to carry the charge through the membrane. However, in the feed solution, the pH was 4.6 and only H_2_PO_4_^−^ ions were present. When H_2_PO_4_^−^ ions entered the membrane, a part of them transformed into doubly and triply charged anions; the H^+^ ions were released in this transformation and returned to the depleted diffusion layer. Thus, the phosphorus total flux, *j*_P_ (equal to the sum of the fluxes of all phosphorus-bearing species) was limited by the H_2_PO_4_^−^ transport from the bulk of feed solution to the membrane surface. The value of *j*_P_ was close to *i*_lim_^Lev^/*F* (*F* is the Faraday constant). A slight excess of *j*_P_ over *i*_lim_^Lev^/*F* was observed, which is due to the electroconvection and exaltation effects. The visualization showed that electroconvection in the studied systems was essentially weaker than in systems with strong electrolytes, such as NaCl.

## 1. Introduction

Ampholytes are substances which have chemical structures and electrical charges that depend on the pH of the medium due to their participation in protonation–deprotonation reactions. Ampholytes comprise a large number of substances, including nutrients or valuable components of food. Among them there are peptides, amino acids, anthocyanins, orthophosphoric, tartaric, citric acid anions, etc. Electrophoresis and electrodialysis with ion-exchange membranes (IEMs) are used increasingly to extract these substances from wastewater [1,2,3,4], products of biomass processing [5], as well as liquid wastes in the food industry [6,7,8,9,10,11].

The attractiveness of these methods is conditioned by the possibility of using not only the difference in particle mobility, but also the ability to change the sign and magnitude of their electric charge when adjusting the pH. The peculiarity of electrodialysis processes is that the composition of the ampholyte-containing solution changes in space and time not only quantitatively, as in the case of strong electrolytes, but also qualitatively. Indeed, the electrodialysis of strong electrolytes (for example, NaCl) is accompanied only by an increase or decrease in their concentration [12]. In the case of ampholytes, the transformation of one form into another takes place not only in the solution, which is located in the intermembrane space [13,14], but also inside an IEM [15,16]. In over-limiting current regimes, generation of H^+^ and OH^−^ ions in protonation–deprotonation reactions with the participation of fixed groups at the membrane/depleted solution boundary [17,18] can significantly affect this transformation. These reactions can cause the so-called barrier effect [19,20]. It lies in the fact that a change in the pH of an ampholyte-containing solution at the membrane surface facing the desalination compartment entails the transformation of ampholyte species. Depending on the value of pH change, the ampholyte ions, which migrate from the solution bulk towards the membrane as counterions, can turn into molecular (zwitterionic for amino acid) form or into a co-ion whose charge sign is opposite to the charge sign of the ampholyte particles in the bulk solution. The molecular (zwitterionic) form cannot be transported through the IEM at the same rate as the counterions; the coions are ejected by the electric field from the near-membrane region into the bulk solution. If the generation of H^+^ and OH^−^ ions occurs at both membranes forming the desalination channel, the ampholyte cations, which are formed at the anion-exchange membrane (AEM) are delivered by the electric field to the cation-exchange membrane (CEM), where they are transformed into anions and return back to the AEM, where they change the charge sign again. This phenomenon is called the circulation effect [21]. Both effects are used for the purification of amino acids or carboxylic acids from mineral impurities [22], as well as for the separation of inorganic ampholytes, such as sulfates and phosphates [23].

The transformation of an ampholyte from one form to another is also possible when it enters or leaves the membrane. This is due to the fact that the internal solution of AEM is more alkaline than the external solution; the pH of the internal solution is one or two units higher than in that of the external solution [16,24,25,26]. The reason is that the H^+^ ions are pushed out from an AEM as coions. Similarly, due to the Donnan exclusion of OH^−^ ions from a CEM, the internal solution of this membrane has a pH value 1–2 units lower than in the external solution. As a result of this pH shift, the effects of facilitated diffusion [27,28] and facilitated electromigration [21,29] occur inside IEMs. The essence of these effects lies in the fact that getting into the acidic (or alkaline) medium inside the IEM, an amino acid zwitterion acquires a charge opposite to the charge of the membrane’s fixed sites. After such a transformation, it easily passes through the IEM as a counterion. In our recent work [30], a similar mechanism was described, which explains a relatively high transport of ammonium ions (coions) through an AEM. In this case, the positively charged ammonium ion enters the alkaline AEM medium and transforms into a molecular form, which is not affected by the Donnan exclusion.

Thus, the transport of ampholytes in systems with ion-exchange membranes is coupled with chemical reactions of protonation–deprotonation. The influence of these reactions on the behavior of membrane systems in conditions of applied electric current has been described in theoretical works [31,32]. The dependence of concentration profiles of ampholyte species as well as their transport numbers in the membrane and adjacent diffusion layers on the applied current density was calculated. However, experimental verification of the model predications was not carried out.

As for the transport of phosphoric acid anions through AEM, this subject is of considerable interest not only for theory but also for practice. Isolation, purification, and concentration of these anions from municipal wastewater [33,34,35], animal waste [2,36], and sludge generated after its biological treatment [37,38] not only reduces the anthropogenic impact on the environment, but also allows for obtaining cheap fertilizers with simultaneous production of electricity [38]. It is important that the current efficiency in the recovery of phosphorus from solutions containing H_2_PO_4_^−^ ions is significantly lower as compared to similar processes in the case of nitrates, chlorides, and other ions that do not undergo protonation–deprotonation reactions [2,34,39,40,41]. With an increasing potential drop, the phosphorus recovery efficiency from the desalination compartment first grows rapidly, but then remains unchanged over a wide range of voltages [23]; the recovery efficiency largely depends on the pH of the treated solution [39]. Some researchers explain the low efficiency of phosphorus recovery by steric hindrances that arise when transporting large, highly hydrated phosphoric acid anions [2,34].

Paltrinieri et al. [42], using the material balance equations, came to the conclusion that electric current can be transported through AEMs by doubly charged HPO_4_^2−^ anions, while the feed solution contains only an NaH_2_PO_4_ solution, where only the singly charged H_2_PO_4_^−^ anions are present. Note that, as a rule, when choosing a current mode, researchers are guided by the limiting current, *i*_lim_, which is found from the intersection of the tangents to the initial part and the inclined plateau of current–voltage characteristic (CVC) curves [34,40]. Another way to find *i*_lim_ is through the use of the Cowan–Brown method [43] for CVC processing [44].

In this work, we report the experimental CVC and partial current densities of all orthophosphoric acid anions, namely, the H_2_PO_4_^−^, HPO_4_^2−^, and PO_4_^3−^ ions, through Neosepta AMX and Fujifilm AEM Type X membranes in the case of a 0.02 M NaH_2_PO_4_ feed solution. As well, we found the fluxes of phosphorus-bearing species through the membranes under study and the partial currents of H_2_PO_4_^−^ and H^+^ ions in the depleted diffusion layer adjacent to the membrane. We compared all these characteristics with the results of a simulation made using the model developed earlier [31,32]. We showed that the phosphorus-bearing species through an AEM in the ED process was less than one would expect if judging by the conventional treatment of CVC. Accordingly, we focused on the determination of limiting current density using the tangent intersection method and the Cowan–Brown method.

## 2. Results and Discussion

### 2.1. Total and Partial Current–Voltage Characteristics

Figure 1 and Figure 2 show the experimental and calculated total and partial current-voltage characteristics (CVCs) of an AMX membrane in NaCl (Figure 1) and NaH_2_PO_4_ (Figure 2) solutions. The calculations are made using a mathematical model developed earlier [31,32] and briefly described below.

In the case of NaCl (Figure 1), the shapes of the total and partial CVCs were close to those described in the literature [45,46] for strong electrolytes, which do not participate in the proton-exchange reactions. The experimental limiting current, which is determined by the tangent intersection method (Figure 1), was close to $i_{\lim}^{Lev}$. Note that the shape of the CVCs in the range from 1.5 to 2.5 mA·cm^−2^ was rather particular. There was a region where the value of *Δφ*′ decreased when the current density increased. The differential resistance of the membrane system within this current range was negative: *R_dif_* = *d*(*Δφ*)/*di* < 0. This “anomaly” is known for the AMX and some other IEMs [47], it is due to the early electroconvective vortex formation at electrically and geometrically heterogeneous surfaces. With growing current density in the indicated range of currents, the increasing electroconvection (occurring as electroosmosis of the first kind [48]) makes the depleted solution resistance lower. Approaching the limiting current density was manifested by a fast increase in the differential resistance with increasing *i*. Thus, to take into account the singularity *R_dif_* < 0 when determining the limiting current density, we drew the tangents to close to the linear segments of the CVCs, just before and after the occurrence of the limiting state, as shown in Figure 1.

The partial current density of OH^−^ ions became noticeable at a reduced potential drop corresponding to the limiting current, then it slowly increased with increasing potential drop in the range corresponding to the CVC plateau (section II) and it sharply increased when *Δφ*′ exceeded 0.8 V, i.e., in the over-limiting regime (section III).

The shape of $i_{OH^{-}}^{AEM}$ versus *Δφ*′ dependence shown in Figure 1 is fully consistent with well-established ideas about the development of water splitting with the catalytic participation of fixed-membrane groups: the generation of H^+^ and OH^−^ ions at the membrane/solution interface begins when the concentration of NaCl in the boundary solution at the membrane surface reaches values of about 5 × 10^−5^ M [49], and the H^+^ (OH^−^) ions in the depleted solution start to compete with the Na^+^ and Cl^−^ ions. This state occurs when the current density is close to the limiting current density at a potential drop across the depleted diffusion layer of about 0.25 V [49]. The rate of water splitting increases with an increase in the potential drop [50]. The partial current of Cl^−^ ions, $i_{Cl^{-}}^{AEM}$, continues to grow after reaching the value $i_{\lim}^{Lev}$ due to the effect of exaltation of the limiting current [51,52] and the development of electroconvection [49,53].

In the case of the NaH_2_PO_4_ solution, the model developed in References [31,32] predicts the presence of two inclined plateaus on the total CVC (Figure 2). In the region of the first plateau (about 0.05 V of the reduced potential drop), the partial current density of singly charged ions H_2_PO_4_^–^ in the membrane reached a maximum value that was close to the limiting current calculated by the Leveque equation. The limiting current calculated by the Leveque equation is the limiting value of the partial current carried by H_2_PO_4_^−^ ions in solution, where this current is limited by diffusion through the depleted diffusion layer. When *Δφ*′ is close to 0.05 V, the concentration of H_2_PO_4_^–^ ions at the AEM surface reach a value which is much lower than the concentration value in the bulk solution, hence, its diffusion flux density attains a maximum [31,32].

Thus, the appearance of the first plateau was due to the saturation of the NaH_2_PO_4_ salt diffusion from the solution to the membrane surface (Figure 2b).

The decrease in NaH_2_PO_4_ concentration in the solution at the AEM surface led to a stronger Donnan exclusion of protons from the membrane. As a result, the pH of the AEM internal solution increased and a higher part of the singly charged phosphate H_2_PO_4_^–^ ions transformed into doubly charged HPO_4_^2–^ ions when crossing the membrane interface: H_2_PO_4_^−^→HPO_4_^2–^ + H^+^ (insert in Figure 2a). The protons released into the solution at the depleted membrane interface were involved in the charge transfer in the depleted DBL forming partial current density *i*^s^_H+_ (Figure 2b). When the fluxes of PO_4_^3−^ and OH^−^ ions in the membrane were negligible, $i_{H^{+}}^{s}=i_{HPO_{4}{}^{2-}}^{AEM}$ (Figure 2a). These protons appearing in solution and the doubly charged HPO_4_^2–^ anions in the membrane caused the rise of current density above the first limiting current density, $i_{\lim}^{1}$. The second plateau (and limiting current, $i_{\lim}^{2}$) was observed when the membrane was completely converted into the HPO_4_^2−^ form. In this state, the flux of protons, released when the H_2_PO_4_^−^ anions entered the membrane and transformed into HPO_4_^2−^, was saturated.

Experimental values of the partial current densities through the AMX and Fuji membranes were calculated using the value of the HPO_4_^2–^ ion flux density, $j_{H_{2}PO_{4}{}^{-}}^{s}$, which was measured using the method described in Section 3.2.1. Namely, $j_{H_{2}PO_{4}{}^{-}}^{s}$ was found from the rate of the NaH_2_PO_4_ removal from the diluate in the batch mode ED, Equation (14). Then, using Equations (15)–(20), it was possible to calculate the partial current densities of the H_2_PO_4_^–^, HPO_4_^2–^, PO_4_^3–^ and OH^−^ ions in a membrane, by applying these equations in pairs in a proper range of pH. For example, when the pH of the membrane internal solution was between 5 and 10, Equations (15) and (18) allow calculation of $i_{H_{2}PO_{4}{}^{-}}^{AEM}$ and $i_{HPO_{4}{}^{2-}}^{AEM}$. If the calculated value of $i_{H_{2}PO_{4}{}^{-}}^{AEM}$ is less than zero, it insinuates that the pH in the membrane is higher than 10, and the *PO_4_^3−^* anions are involved in current transfer instead of H_2_PO_4_^−^. Then, Equations (19) and (20) must be used. As Figure 3 shows, the contribution of the *PO_4_^3−^* ions into the charge transfer in the membranes becomes significant only at relatively high current densities (exceeding approximately 3*i*_lim_^Lev^). Thus, we did not reach the conditions where the OH^−^ ions pass across the membrane because of the limitations of the measuring device.

When HPO_4_^2−^ and PO_4_^3−^ ions cross the membrane boundary with the enriched solution and appear in aqueous solution, they capture H^+^ ions from water, which leads to an increase in pH of the enriched solution. It can be also interpreted as the generation of OH^−^ ions at the membrane/enriched solution interface. Thus, in membrane systems with NaH_2_PO_4_ solution, the process of generation of H^+^ and OH^−^ ions is separated in space: the H^+^ ions appear at the depleted solution/membrane interface and the OH^−^ ions at the membrane/enriched solution interface. The contribution of the OH^−^ transport in the charge transfer in the membrane becomes essential only at sufficiently high current densities/potential drops, when the pH of the internal membrane solution is greater than 13 (see Section 3.2.1.). Therefore, when pH < 13, the transport of OH^−^ ions across the membrane is negligible. In the experiments in this study, the situation where the transport of OH^−^ ions was measurable was not achieved because of the limitations of the measuring device.

#### 2.1.1. Contributions of Electroconvection and Exaltation Effects.

The comparison of the results of the experiment (Figure 3 and Figure 4a) and simulation using the model [31,32] (Figure 2 and Figure 4a) for the values of *i_tot_* and *i_i_* as functions of *Δφ*^′^ showed that they were in relatively good agreement. The difference lies in the fact that for a given reduced potential drop (up to *Δφ*^′^ = 0.2 V, which is the maximum value for which the computations according to the model are possible), the experimental values of *i_tot_* and $i_{H_{2}PO_{4}{}^{-}}^{AEM}$ were slightly higher than that predicted by the model. The experimental phosphorus flux density, $j_{P}=i_{H_{2}PO_{4}{}^{-}}^{s}/F$, through the membranes under study (Figure 4), determined by Equation (13), also exceeded the values calculated using this model [32].

The increase in the flux density of *H_2_PO_4_^−^* ions from the bulk solution to the AEM surface ($j_{H_{2}PO_{4}{}^{-}}^{s}$) over the value corresponding to the limiting current density *i*_lim_^Lev^ can be caused only by two effects: (1) electroconvection, which reduces the effective thickness of the diffusion layer and (2) exaltation of the *H_2_PO_4_^−^* current. As shown in References [49,54], the contribution of gravitational convection in such systems is negligible. Figure 2b shows the calculated partial current density of *H_2_PO_4_^−^* ions, ${\left(i_{H_{2}PO_{4}{}^{-}}^{s}\right)}_{electrodif}$, which would occur if the ion transport was only due to electro-diffusion, taking into account the effect of exaltation [51], while the electroconvection was not taken into account. The calculation is carried out using the following equation [32]:(1)$${\left(i_{H_{2}PO_{4}{}^{-}}^{s}\right)}_{electrodif}=i_{\lim}^{Lev}+i^{ex}=i_{\lim}^{Lev}+\frac{D_{H_{2}PO_{4}^{-}}}{D_{H^{+}}}i_{H^{+}}$$
where $i_{\lim}^{Lev}$ is the limiting value of the *H_2_PO_4_^−^* ion current density (calculated by Equation (6)) achieved by electro-diffusion in the absence of exaltation, and the second term, *i^ex^*, describes the increase in the current density owing to the exaltation effect. This effect is due to the protons released from the *H_2_PO_4_^−^* ions when they enter the membrane. The H^+^ ions carry a positive charge, which creates an additional electrostatic field attracting the *H_2_PO_4_^−^* anions from the bulk solution to the depleted membrane surface [51].

However, as can be seen from Figure 2b, the exaltation effect is too weak to provide the experimentally observed *H_2_PO_4_^−^* ions’ flux in the diffusion layer near the membrane surface (equal to the total flux of phosphorus atoms through the membrane). Consequently, the main cause of growth should be electroconvection, which, as it known from References [48,49,53,55], significantly increases the mass transfer in dilute solutions of strong electrolytes. Our experiments show that in the systems under study, in the case of NaCl solution, the first large vortex structures (about 50 μm in size) are visualized at the outlet of the channel under study, when *i/i*_lim_^Lev^ = 2. With an increasing current, the vortex structures gradually spread along the entire length of the channel, and the dimensions of the vortices increase. At *i*/*i*_lim_^Lev^ = 6 (Figure 5a), the vortex structures occupy almost half of the 900 μm intermembrane space.

The scenario of the electroconvective structures’ development in the NaH_2_PO_4_ solution was the same as in the case of the NaCl solution. The difference was that the vortices of the same diameter emerged at essentially higher current densities, e.g., those with a diameter of 50 μm only formed when *i/i*_lim_^Lev^ = 6 (Figure 6b).

The reason for the weaker electroconvection in the case of the NaH_2_PO_4_ solution, apparently, was the high rate of H^+^ ion generation at the same value of the *i/i*_lim_ ratio. Getting into the space charge region at the depleted membrane surface, the H^+^ ions caused a decrease in its density [56]. In the case of strong electrolyte solutions, the generation of H^+^ (and OH^−^) ions occurred, as noted above, after reaching a certain threshold concentration of salt ions near the membrane surface. After overcoming this threshold (which requires a certain minimum potential drop), the H^+^ and OH^−^ ions were generated as a result of “water splitting” occurring as proton-exchange reactions with catalytic participation of fixed groups at the membrane/solution interface [57,58,59,60]. In the case of the NaH_2_PO_4_ solution, the generation of H^+^ ions occurred in the non-threshold mode as a result of the dissociation of a part of H_2_PO_4_^−^ ions when they enter the membrane bulk. This process takes place as soon as an external electric field was applied to the membrane.

#### 2.1.2. Rate of Phosphorus Removal from Feed Solution and Its Theoretical Assessment.

The rate of phosphorus removal from the feed solution is controlled by the flux density of the H_2_PO_4_^−^ ions from the feed bulk solution to the anion-exchange membrane surface. Since the pH of the feed solution was close to 4.5, there were no other phosphorus-bearing species except for the H_2_PO_4_^−^ ions. As mentioned above, the electro-diffusion flux of the H_2_PO_4_^−^ ions was limited by the first limiting current density, *i*_lim_^1^ (which was very close to the theoretical value *i*_lim_^Lev^); exaltation and electroconvection can enhance this transport.

It should be noted that the charge transfer by doubly and triply charged anions of orthophosphoric acid through the membrane, as well as the transport of OH^−^ ions, are parasitic processes if the purpose of electrodialysis is the removal of phosphorus from the feed solution. Therefore, the phosphorus flux density, *j*_P_, which is transported through the AEM by all the phosphorus-bearing particles, Equation (13), turns out to be significantly less than what one would expect if the formula *j = i_lim_*/*F* (where *j* is the effective transport; *i_lim_* is the limiting current density through the membrane) is used by the analogy with strong electrolytes. In the practice of electrodialysis, the value of *i*_lim_ was found using CVC and applying the tangent intersection method [12], or the Cowan–Brown method [43], as explained above. In the last case, the integral resistance of the membrane system (equal to ∆*φ**/i*) was presented as a function of the inverse current density, 1/*i* (Figure 4).

As can be seen in Figure 5, the experimental determination of *i*_lim_^1^ was difficult if the CVC with the total potential drop, ∆*φ*, was used, both when the tangent intersection method or the Cowan–Brown method were applied. However, the second plateau and the second limiting current density, *i*_lim_^2^, which corresponds to the state where the membrane is completely transformed into the form of doubly charged *HPO_4_^2−^* ions, were essentially better pronounced on the CVC. When the Cowan–Brown coordinates (Figure 5a,b, insertions) were used, the region of the curve related to *i*_lim_^1^ was very close to a straight line. The limiting current density $i_{\lim}^{C-B}$, usually found as the intersection point to two nearly linear regions of the ∆*φ*/*i* versus 1/*i* curve, was then rather close to the second limiting current, *i*_lim_^2^, since the singularity of the curve related to *i*_lim_^1^ in these coordinates was quasi invisible, especially if a more rough scale (as usual) was used. Thus, when the CVC is recorded with insufficient accuracy, confusion may occur: the second limiting current density can be taken as the current density, which determines the rate of phosphorus removal.

The first limiting current is difficult to see, apparently, because the approach to saturation in the *H_2_PO_4_^−^* ion diffusion transport in solution is accompanied by a rapid increase in the contribution to charge transfer of H^+^ ions, released when *H_2_PO_4_^−^* enters the membrane. As shown by mathematical modeling [32], the plateau of the first limiting current was the less noticeable the smaller the difference in values of the dissociation constants related to the first (K_a1_) and the second (K_a2_) steps of the ampholyte dissociation. In the case of phosphoric acid, the difference between pK_a1_ = 2.12 and pK_a2_ = 7.21 was nevertheless rather large. This difference was essentially lower for a number of other weak acids, such as tartaric acid (pK_a1_ = 2.98 and pK_a2_ = 4.34 [61,62]). Thus, one can expect that the detection of the first limiting current would be even more difficult than in the case of phosphoric acid.

The practical question of how to detect the first limiting current density is quite important. Let us note that the plateau related to *i*_lim_^1^ was easily detectable when using the *i* versus *∆φ*′ coordinates (Figure 3a,b). The use of the *i* versus *∆φ* coordinates (Figure 5a,b) makes the detection of *i*_lim_^1^ essentially more uncertain; the second limiting current may be taken for the first one. Nevertheless, if the maximum rate of phosphorus removal, *j*_P_^max^, is evaluated by *i*_lim_^2^, the obtained value will be approximately 1.5 times higher than the experimentally measured value of *j*_P_^max^ in this study. Apparently, it is particularly the overestimation of the limiting current, which is responsible for the unexpectedly high energy consumption and low current efficiency of the ED recovery of phosphorus and other ampholyte species, reported by several authors [10,34,39,41].

## 3. Materials and Methods

### 3.1. Membrane and Solutions

The Neosepta AMX anion-exchange membrane was manufactured (Astom, Japan) using a previously described method [63,64,65]. This membrane contained quaternary ammonium bases and a small amount of secondary and tertiary amines [66]. It had an undulated surface: there were “hills” and “valleys” located in staggered order. This order was caused by the geometry of PVC reinforcing fabric [67]. The average difference between the highest and the lowest points on the surface of the swollen membrane was about 30 μm [67]. The characteristic size of the geometric inhomogeneity along the surface (280 μm) was close in magnitude to the thickness of the diffusion layer adjacent to the membrane in the compartments of the electrodialysis cell used in the study. The AMX membrane thickness in a 0.02 M NaCl solution and the ion exchange capacity of swollen membrane were equal to 125 ± 5 μm and 1.23 ± 0.05 mmol·g^−1^ wet, respectively.

The basis of the homogeneous AEM Type X membrane (Fujifilm, The Netherlands) was a three-dimensional structure (substrate) of inert polyolefin fibers [68], which were obtained by electroforming [69] methods. The aerogel formed by the fibers was pressed to a predetermined thickness. The space among the fibers was then filled with aliphatic polyacrylamide [70,71] and functionalized with quaternary ammonium bases [72]. The AEM Type X thickness in a 0.02 M NaCl solution and the ion exchange capacity of swollen membrane were equal to 120 ± 5 μm and 1.50 ± 0.05 mmol·g^−1^ wet, respectively.

All membrane samples underwent a standard salt pretreatment [73] and then were equilibrated with a 0.02 M salt solution before experiments. The solutions of sodium chloride (NaCl) and monosodium phosphate (NaH_2_PO_4_) were prepared from a crystalline salt (analytical grade) provided by OJSC Vekton; the 0.10 M NaOH solution was prepared from a titrant (manufactured by Uralkhiminvest, Russia). NaOH was used to maintain a constant pH value of the solution circulating through the compartments of the measuring cell. Distilled water, with an electrical conductivity of 0.8 µS·cm^−1^, pH = 6.0 ± 0.2, and temperature of 25 °C, was used to prepare the solutions. Table 1 shows several characteristics of the electrolyte solutions used in the experiments.

Figure 7 shows the distribution of the phosphoric acid species (in mole fractions) as a function of pH. It is calculated using the equations for the equilibrium of protonation–deprotonation reactions on the first, second, and third steps:(2)$$H_{3}{\mathrm{PO}}_{4}+H_{2}O\Leftrightarrow H_{2}{\mathrm{PO}}_{4}^{-}+H_{3}O^{+}$$
(3)$$H_{2}{\mathrm{PO}}_{4}^{-}+H_{2}O\Leftrightarrow {\mathrm{HPO}}_{4}^{2-}{+H}_{3}O^{+}$$
(4)$${\mathrm{HPO}}_{4}^{2-}+H_{2}O\Leftrightarrow {\mathrm{PO}}_{4}^{3-}+H_{3}O^{+}$$

Negative logarithms of the equilibrium constants of these reactions at 25 °C are equal to [61] 2.12 (pK_a1_), 7.21 (pK_a2_), and 12.34 (pK_a3_).

### 3.2. Methods

The measurements of the electrochemical characteristics of AEMs were carried out at a temperature of 25 ± 1 °C using a flow-through four-compartment electrodialysis laboratory cell connected to an Autolab PGSTAT-100 electrochemical complex. The setup and the cell are described in detail in Reference [30,49]. A schematic design of the setup is shown in Figure 8. The intermembrane distance in the desalination compartment (14), *h*, was equal to 6.6 mm; the linear flow velocity of the electrolyte solution through each chamber, *V*, was 0.4 cm·s^−1^; the polarizable area of the membrane was 2 × 2 cm^2^; the tips of Luggin’s capillaries (5) used to record the potential drop across the membrane under study were located at a distance of approximately 0.8 mm from its surfaces. The plexiglass frames separating the membranes in the electrodialysis cell (Figure 7) were equipped with special guides in the shape of a comb, which provided laminar regime of the solution flow in the cell compartments.

The current–voltage characteristics (CVC) were measured in the galvanodynamic mode at a current sweep rate of 0.02 μA·s^−1^. Measuring Ag/AgCl electrodes EVL-1M3.1 (Gomel, Belorussia) with a working area of several tenths of cm^2^ immersed in saturated KCl solution were used. The volume of the feed solution in tank (1) at the beginning of the experiment was 5 dm^3^. This solution was fed to all of the compartments of the cell and then returned to the same tank. Due to the relatively large volume of this solution and the fact that the diluate and concentrate were mixed before returning to tank (1), the deviation of the species’ concentrations in the tank from their initial values during one run of the experiment did not exceed 1%.

The total potential drop, ∆*φ*, measured using Luggin capillaries (5) depends, along with the potential drop across the polarized diffusion layers, on the resistance of the membrane and solution. The latter is a function of the distance between the membrane and capillaries (5) [76]. This distance is difficult to find and reproduce when replacing one membrane with another one. To exclude this difficulty, the corrected potential drop ∆*φ*′ [77] is used instead of ∆*φ*:(5)$$\Delta \phi^{\prime }=\Delta \phi -iR_{ef}$$
where the effective resistance of the membrane system *R_ef_* (Ohm cm^2^) includes the ohmic resistance of the space (membrane + solution) among the measuring capillaries, as well as the diffusion resistance of the interphase boundaries of depleted and enriched diffusion layers [49]. The value of *R_ef_* is found from the initial part of the CVC by extrapolation *i*→0 in the coordinates *i* versus *d*(∆*φ*) *di.*

The limiting current density, $i_{\lim}^{Lev}$, was calculated using the Leveque equation obtained in the framework of the convective–diffusion model [78]. A 1:1 electrolyte can be expressed as:(6)$$i_{\lim}^{Lev}=1.47\left[\frac{FDC}{h(T_{i}-t_{i})}{\left(\frac{h^{2}V}{LD}\right)}^{1/3}-0.2\right]$$
where *L* is the desalination channel length; *C* (mole·dm^−3^) is the electrolyte concentration of the feed solution at the entrance to the desalination channel; *D* is the electrolyte diffusion coefficient at infinite dilution (Table 1); *t_i_* is the transport number of salt counter-ion in solution at infinite dilution (Table 1); *T_i_* is the effective transport number of salt counter-ion in the membrane. The latter is defined as the fraction of the electric current transferred by ions “*i*” without restrictions on the concentration or pressure gradients; for the investigated membranes in the given solutions, the values of *T_i_* for the salt coions (Na^+^) are taken to be zero.

According to the definition above:(7)$$T_{i}=\frac{z_{i}j_{i}F}{i}=\frac{i_{i}}{i}$$
where *z_i_* and *j_i_* are the ion charge number and the flux density, *F* is the Faraday constant; $i_{i}=z_{i}j_{i}F$ is the partial current density of ion *i*.

Since *j_i_* in general cases change with the coordinate normal to the membrane surface, *T*_i_ is a function of this coordinate. In particular, if the solution pH is close to 4.5, the main phosphorus-bearing species is H_2_PO_4_^−^. In the membrane, pH is 1–3 units higher [16,24,26], therefore, together with H_2_PO_4_^−^, the presence of doubly charged HPO_4_^2−^, and even triply charged PO_4_^3−^, is possible.

The average of the cell length thickness of the diffusion layer, *δ*, is estimated by combining the Leveque and the Peers equations [32]
(8)$$\delta =0.68h{\left(\frac{LD}{h^{2}V}\right)}^{1/3}$$

The values of *i_lim_^Lev^* and *δ* for each electrolyte are summarized in Table 1.

#### 3.2.1. Effective Transport Numbers, T_i_

The values of the effective transport numbers, *T_i_*, and partial currents of counter-ions are obtained using the same cell as when measuring the total CVC. For measuring *T_i_*, desalination compartment (14) was fed with a solution from the additional tank (2). Before an experiment was run, the circuit consisting of the desalination channel, tank (2), and connecting tubes were filled with a 0.035 M solution of the electrolyte under study (NaCl or NaH_2_PO_4_). The other circuits of the cell were fed with a 0.02 M solution of NaCl or NaH_2_PO_4_ from tank (1). The initial electrolyte concentration in tank (2) was 0.035 M; it decreased during one experimental run to 0.01 M. In this concentration range, the coion (Na^+^) transferred through the anion-exchange membrane can be neglected [79]. As well, in this concentration range, the concentration of carbonic acid dissociation products formed due to the continuous dissolution of atmospheric carbon dioxide in the working solution is negligible [80].

The initial volume of the solution in the desalination circuit (including tank (2), the desalination compartment and connecting tubes) was 0.100 ± 0.002 dm^3^; the solution flows among the membranes at a velocity of 0.40 ± 0.01 cm·s^−1^. The volume, concentration, and flow velocity of the solution were selected in a way to ensure quasi-stationary conditions for desalination during an experimental run. In separate experiments, it was found that to meet these conditions, the rate of electrical conductivity decreased in the desalting circuit (in the solution circulating through tank (2)) should not exceed 1% per minute [80].

The experiment was carried out at a constant corrected potential drop ∆*φ*′. At equal time intervals (10 min), the electrical conductivity (*κ*), pH, and temperature of the solution in tank (2) (Figure 8) were recorded using a combined electrode for pH measurement (11) connected to the pH meter Expert 001 and the submersible conductometric cell (12) connected to the conductometer Expert 002 (LLC “Ekoniks expert”, Russia). In all cases presented in this paper, the solution leaving the desalination compartment of the cell (compartment (14) in Figure 8) was acidified. To maintain a constant pH of the desalination stream close to 5.7 ± 0.05 (NaCl) or to 4.6 ± 0.05 (NaH_2_PO_4_), a 0.1 M NaOH solution was continuously added into tank (2) using a micro capillary (13). The electrolyte concentration in tank (2) was periodically determined by measuring the solution conductivity in this tank and using an equation, which connects the concentration of NaCl or NaH_2_PO_4_ solution with its conductivity.

The concentration of the solution in the desalination circuit changes with time due to the transfer of electrolyte ions through the membranes forming the desalination compartment to the neighboring compartments, as well as due to the addition of a titrant into this tank. The mass balance for a salt ion 1 (e.g., a cation) in the desalination circuit is described by Equation (9), under the assumption that the difference in concentrations in the different elements of this circuit (tank (2), the cell and connecting tubes) is negligible [30]:(9)$$\bar{V}\frac{dC}{dt}=-\frac{i(T_{1}^{CEM}-T_{1}^{AEM})S\;n}{z_{1}F}+C_{T}\frac{d\bar{V_{T}}}{dt}$$
where *i* is the current density; $T_{1}^{CEM}$ and $T_{1}^{AEM}$ are the effective transport numbers of ion 1 in the cation-exchange membrane and anion-exchange membrane, respectively, which form the desalination compartment; *C* is the current salt concentration (NaCl or NaH_2_PO_4_) in tank (2); $\bar{V}$ is the volume of the solution in the desalination circuit; *n* is the number of desalination compartments (*n* = 1); *C_T_* and ${\bar{V}}_{T}$ are the concentration and volume of the titrant, which is added to tank (2) to compensate for changes in pH caused by the H^+^ (OH^−^) generation at the membrane/solution interfaces; *S* is the area of active (polarizable) membrane surface. As it was mentioned above, the solution after passing the desalination compartment (14) was acidified, hence, a NaOH solution was added to the desalination stream.

The first term on the right-hand side of Equation (9) describes the decrease in the salt concentration in the tank (2) caused by the flow of electric current across the AEM and CEM; the second term describes the addition of a titrant in tank (2).

Consider the case of NaCl solution and a CEM under study. If the transfer of cations (Na^+^) through the auxiliary AEM is neglected ($T_{1}^{AEM}$ = 0), it follows from Equation (9) that:(10)$$j_{Na^{+}}^{CEM}=\frac{i_{Na^{+}}^{CEM}}{F}=\frac{iT_{Na^{+}}^{CEM}}{F}\approx -\frac{\bar{V}}{S}\frac{dC}{dt}+\frac{C_{T}}{S}\frac{d\bar{V_{T}}}{dt}$$

In the case where an AEM is under study, Equation (9) should be written for the Cl^−^ ions. Like as above, we assumed that the auxiliary membrane (CEM) was not permeable for anions, i.e., $T_{Cl^{-}}^{CEM}$ = 0. Since the Cl^−^ ions were not present in the titrant added to tank (2), Equation (9) reads:(11)$$j_{Cl^{-}}^{AEM}=\frac{i_{Cl^{-}}^{AEM}}{F}=\frac{iT_{Cl^{-}}^{AEM}}{F}\approx -\frac{\bar{V}}{S}\frac{dC}{dt}$$

When the transport numbers of the salt cation in the CEM (Na^+^) and the salt anion (Cl^−^) in the AEM were determined, the transport numbers of the H^+^ ion in the CEM and OH^−^ ion in the AEM were found according to Equation (12):(12)$$T_{H^{+}}^{CEM}=1-T_{Na^{+}}^{CEM},\;T_{OH^{-}}^{AEM}=1-T_{Cl^{-}}^{AEM}$$

In the case of the NaH_2_PO_4_ solution desalination, the calculation of the partial currents of sodium ions and protons in CEM was carried out according to Equations (10) and (12).

The determination of the partial currents of *H_2_PO_4_^−^*, *HPO_4_^2−^*, *PO_4_^3−^*, and *OH^−^* ions through an AEM was carried out using the material balance equations presented below. The following assumptions were made for their deduction.

1. The total phosphorus flux through an AEM, $j_{P}^{AEM}$, is equal to the sum of fluxes of all phosphorus-bearing ions entering the AEM from the diluate compartment. Since only the *H_2_PO_4_^−^* ions were present at pH = 4.6 in this compartment, we have, $j_{H_{2}PO_{4}{}^{-}}^{s}$:(13)$$j_{H_{2}PO_{4}{}^{-}}^{s}=j_{P}^{AEM}=j_{H_{2}PO_{4}{}^{-}}^{AEM}+j_{HPO_{4}{}^{2-}}^{AEM}+j_{PO_{4}{}^{3-}}^{AEM}=\frac{i_{H_{2}PO_{4}{}^{-}}^{AEM}}{z_{H_{2}PO_{4}{}^{-}}F}+\frac{i_{HPO_{4}{}^{2-}}^{AEM}}{z_{HPO_{4}{}^{2-}}F}+\frac{i_{PO_{4}{}^{3-}}^{AEM}}{z_{PO_{4}{}^{3-}}F}$$
where superscripts “*s*” and “*AEM*” relate to the solution at the diluate side of the membrane and the AEM, respectively.

The $j_{H_{2}PO_{4}{}^{-}}^{s}$ value can be determined experimentally by the rate of *NaH_2_PO_4_* concentration decrease in the desalination stream (where the pH was kept constant), in accordance with an equation which is similar to Equation (11):(14)$$j_{H_{2}PO_{4}{}^{-}}^{s}=-\frac{\bar{V}}{S}\frac{dC_{NaH_{2}PO_{4}}}{dt}$$

2. From the distribution of the phosphoric acid species as a function of pH (Figure 8), it follows that only two sorts of phosphorous-containing species can be present in an AEM: either *HPO_4_^2−^* and *HPO_4_^2−^* (when the pH of the internal solution is in the range from 5 to 10) or *HPO_4_^2−^* and *PO_4_^3−^* (pH from 10 to 13). The concentrations of the remaining species in any of the three pH ranges presented above are so small that they can be neglected. The third possible pair, *PO_4_^3−^* and *OH^−^* can coexist at pH > 13. At pH = 13, the molar fraction of *PO_4_^3−^* in the membrane is close to 0.8 and that of *HPO_4_^2−^*, 0.2, while the concentration of *OH^−^* ions is 0.1 M, which is about 5% of the exchange capacity, hence, the total concentration of phosphate species. However, taking into account the high mobility of *OH^−^* ions, it can be assumed that these ions will compete with the *PO_4_^3−^* ions, whose content is dominant at this pH value.

Since the H^+^ ions being coions are excluded from the anion-exchange membrane, the pH of the membrane internal solution is 1 to 2 pH units higher than the pH of the external solution [16]. The coion exclusion increases with the dilatation of the external solution contacting the membrane, as it follows from the well-known Donnan equation [81]. With an increasing current density, the solution contacting the membrane surface at its diluate side becomes increasing diluted and the concentration of the H^+^ ions in the near-surface membrane layer decreases, so that the pH of the internal solution in this layer increases. Therefore, the composition of the membrane near-surface layer on the diluate side varies with the current density: at relatively low current density (low concentration polarization), the pH in the membrane internal solution is relatively low, and this layer contains mainly ions *H_2_PO_4_^−^* and *HPO_4_^2−^*. With increasing current density, the concentration polarization increases, the external boundary solution becomes more diluted, the pH in the membrane internal solution becomes higher, and the membrane near-surface layer is enriched firstly with the *PO_4_^3−^* ions, and then, with the *OH^−^* ions.

In the pH range from 5 to 10 (relatively low current densities), where only *H_2_PO_4_^−^* and *HPO_4_^2−^* are present in the membrane, the total current density, *i*, is carried exclusively by these ions:(15)$$i_{H_{2}PO_{4}{}^{-}}^{AEM}+i_{HPO_{4}{}^{2-}}^{AEM}=i$$
and, according to Equation (13):(16)$$j_{H_{2}PO_{4}{}^{-}}^{s}=\frac{i_{H_{2}PO_{4}{}^{-}}^{AEM}}{F}+\frac{i_{HPO_{4}{}^{2-}}^{AEM}}{2F}$$
where coefficient “2” in the denominator of the second term on the right-hand side stands for $z_{HPO_{4}{}^{2-}}$. Then, the partial current densities of singly and doubly charged phosphorus-bearing anions can be found from the two following equations:(17)$$i_{H_{2}PO_{4}{}^{-}}^{AEM}=2Fj_{H_{2}PO_{4}{}^{-}}^{s}-i$$
(18)$$i_{HPO_{4}{}^{2-}}^{AEM}=2\left(i-Fj_{H_{2}PO_{4}^{-}}^{s}\right)$$

Thus, we can calculate the partial current densities of the *H_2_PO_4_^−^* and *HPO_4_^2−^* species, if we measure the current density, *i*, and the rate of the diluate desalination; the latter allows for the calculation of $j_{H_{2}PO_{4}^{-}}^{s}$, Equation (14).

In the range of pH from 10 to 13.5 (high current densities), where $i_{HPO_{4}{}^{2-}}^{AEM}+i_{PO_{4}{}^{3-}}^{AEM}=i$, in a similar way, we find:(19)$$i_{HPO_{4}{}^{2-}}^{AOM}=2\left(3\;Fj_{H_{2}PO_{4}{}^{-}}^{s}-i\right)$$
(20)$$i_{PO_{4}{}^{3-}}^{AOM}=3\left(i-2\;Fj_{H_{2}PO_{4}{}^{-}}^{s}\right)$$

The coefficients 2 and 3 in Equations (17)–(20) stands for the charge numbers $z_{HPO_{4}{}^{2-}}$ and $z_{PO_{4}{}^{3-}}$, respectively.

At even higher current densities, where all doubly charged orthophosphoric acid anions transform into triple-charged ones ($T_{HPO_{4}^{2-}}=i_{HPO_{4}^{2-}}/i=0$)and the pH of the internal solution exceeds 13.5, the current is transported through the membrane by PO_4_^3−^ and OH^−^ ions. The partial currents of these ions are:(21)$$i_{PO_{4}{}^{3-}}^{AEM}=3\;Fj_{H_{2}PO_{4}{}^{-}}^{s}$$
(22)$$i_{OH^{-}}^{AEM}=i-i_{PO_{4}{}^{3-}}^{AEM}$$

In accordance with Equations (2)–(4) and (13), at the AEM/solution interface, the partial flux of protons entering the depleted diffusion layer is:(23)$$j_{H^{+}}^{s}=j_{HPO_{4}^{2-}}^{AEM}+2j_{PO^{3-}}^{AEM}$$

Taking into account that $i_{k}=j_{k}z_{k}F$, the partial current density of H^+^ ions in the depleted solution near the membrane surface is:(24)$$i_{H^{+}}^{s}=\frac{i_{HPO_{4}{}^{2-}}^{AEM}}{2}+\frac{2i_{PO_{4}{}^{3-}}^{AEM}}{3}$$

#### 3.2.2. Visualization of Electroconvective Vortices.

Visualization of vortex flows near an AMX membrane surface facing the desalination compartment was carried out using a technique similar to that described in Reference [82].

Two poly(methyl methacrylate) frames clung close to the ion-exchange membrane and formed a channel, with a 0.9 mm width and a 3 mm length. A 0.01 M NaCl or 0.01 M NaH_2_PO_4_ solution entered the desalination compartment through the holes in the clamping plates, in which the electrode chambers were located. The linear flow rate of the solution was 0.001 cm·s^−1^. The distance from the investigated membranes to the measuring electrodes was 800 μm. To visualize vortexes, 10 μM rhodamine 6G was added to the solution. Rhodamine 6G is able to fluoresce in a narrow range of wavelengths and it is a large cation. The chlorine or hydrogen phosphate ions were the anions in the system under study. A SOPTOP CX40M optical microscope (China) with a 5× objective and a digital eyepiece camera were used to record these vortexes. The resolution of the digital optical system allows for the recording of the appearance of vortices with a diameter of 20 microns or greater. Digital video recording was carried out simultaneously while measuring the potential difference over the membrane, and the current density through the membrane was set as a function of time. The limiting current density was estimated from the current–voltage characteristic of the membrane using the tangent intersection method.

## 4. Theory

The calculation of the CVC, as well as the transport numbers, partial fluxes, and partial currents of ions in the membrane system was carried out using a stationary 1D model described in detail in References [31,32]. A three-layer system under direct current conditions was considered. It consisted of an anion-exchange membrane (AMX) and two adjacent diffusion boundary layers (DBLs). Migration and diffusion transport of neutral and negatively charged ampholyte species, as well as the Na^+^, H^+^, and OH^−^ ions in all three layers, is described using the Nernst–Planck equation under the local electroneutrality condition. The model assumes the independence of the current density and the sum of the fluxes of all phosphorus-bearing species from the coordinate. As well, local chemical equilibria were assumed among the ampholyte species involved in protonation–deprotonation reactions in the AEM and DBLs. In addition, the condition of the ion-exchange equilibrium (expressed by the Donnan equations) at the membrane/solution boundaries was applied. The ion diffusion coefficients in the solution (Table 1) were taken at the infinite dilution [83], because at the near-limiting current densities, the ion concentrations in the solution next to membrane surface were close to zero. The diffusion coefficients in the membrane were determined from the AMX membrane conductivity [84].

## 5. Conclusions

The transport of the singly charged phosphoric acid anions, H_2_PO_4_^−^, across an AEM was accompanied by dissociation of a part of these anions when entering the membrane. The H^+^ ions released in this dissociation returned to the depleted solution and participates in carrying electric charge. The generation of H^+^ ions occurred without voltage threshold, which existed in the case of strong electrolytes. Depending on the current density, the H_2_PO_4_^−^ anions could transform into HPO_4_^2−^ ions (at relatively low current densities, approximately at *i* < 2.5 *i*_lim_^Lev^) or PO_4_^3−^ ions (at higher current densities). At relatively high current densities/voltages along with this transformation, water splitting with catalytic participation of membrane functional groups can occur.

The dissociation of the H_2_PO_4_^−^ anion occurred because the pH of the membrane internal solution was higher than the pH of the external solution. The shift in pH was due to the Donnan exclusion of the H^+^ ions, which were coions for the AEM. The degree of the H^+^ exclusion (hence, the pH of the internal solution) depends on the electrolyte concentration in the boundary solution: the more dilute this solution, the higher the internal solution’s pH. Since the concentration of the boundary solution decreased with increasing current density, the internal solution pH increased with increasing *i*, as a consequence, the equivalent fraction of doubly and triply charged anions in the membrane increased.

Each step of the H_2_PO_4_^−^ anion dissociation had a response on the I–V curve. When the boundary electrolyte concentration became much lower than the bulk concentration, an inclined plateau of the first limiting electric current was observed. When *i > i*_lim_^1^, a more complete transformation of the H_2_PO_4_^−^ anion into the HPO_4_^2−^ ions in the membrane occurred, the H^+^ ions liberated in this reaction gave rise to the electric current. The second plateau appeared when the membrane passed completely in the HPO_4_^2−^ form. The next rise in the current density occurred due to the appearance of the PO_4_^3−^ ions in the membrane and a new portion of the H^+^ ions ejected into the depleted solution. When HPO_4_^2−^ and PO_4_^3−^ ions crossed the membrane boundary and into the enriched solution on the other side of the membrane, they captured H^+^ ions from water, which led to an increase of pH in the enriched solution. It can be also interpreted as generation of OH^−^ ions at the membrane/enriched solution interface. Thus, in AEM/NaH_2_PO_4_ systems, the process of generation of H^+^ and OH^−^ ions is separated in space: H^+^ ions appear at the depleted solution/membrane interface and OH^−^ ions at the membrane/enriched solution interface.

Following from the above, phosphorus can be transferred across the membrane by singly, doubly or triply charged anions. However, the origin of all these species are the H_2_PO_4_^−^ anions presented in the feed solution. Hence, the sum of the phosphorus-bearing species fluxes is equal to the flux of the H_2_PO_4_^−^ anions from the bulk solution to the depleted membrane boundary. The H_2_PO_4_^−^ ions are transported by electro-diffusion including the exaltation effect and by electroconvection. The Leveque equation gives the limiting current density not including the exaltation effect; the latter gives an increase in the phosphorus flux approximately equal to 10%, and electroconvection (in the conditions of our experiment) gives the contribution of about 60% to the overall flux. The occurrence of electroconvection was established via visualization of electroconvective vortices. It was shown that under the same value of *i*/*i*_lim_^Lev^, the size of electroconvective vortices in the case of 0.02 M NaH_2_PO_4_ solution was essentially lower than in the case of NaCl solution of the same concentration. Thus, what is important is that increasing current densities/voltages in ED systems does not lead to expected growth of the extraction degree of phosphorus. Increasing current density is spent particularly in the H^+^ ions’ transport. Electroconvection is less effective, as in the case of NaCl, since the H^+^ ions reduce the space charge at the depleted solution/membrane interface. Hence, with increasing current density over *i*_lim_^Lev^, the current efficiency of the phosphorus recovery strongly decreases.

Another interesting remark is that the conventional application of the Cowan–Brown method to find the limiting current density, gives the second limiting current density. This current density is nearly two times higher than *i*_lim_^1^, which mainly determines the flux of phosphorus across the membrane.

## Figures and Tables

**Figure 1 ijms-20-03593-f001:**
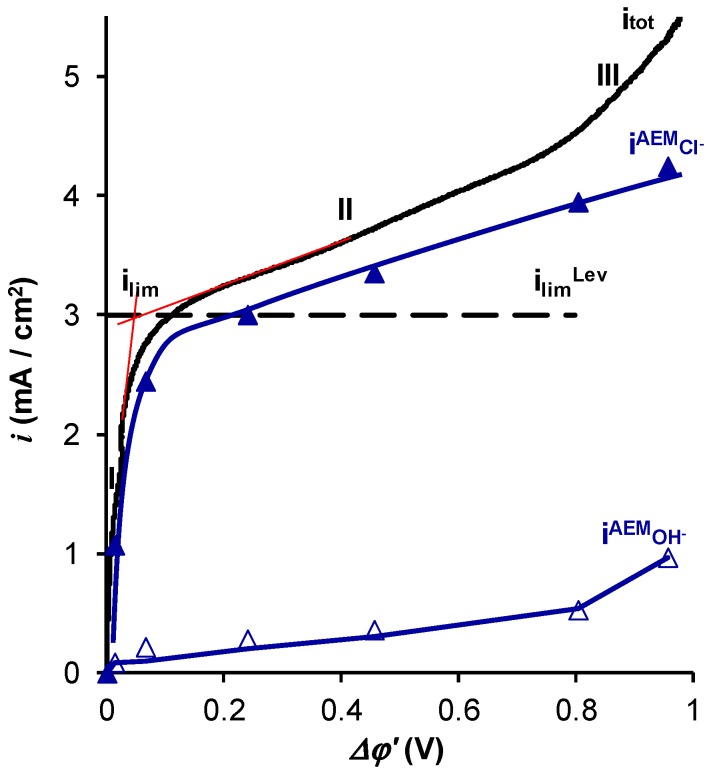
Total and partial current–voltage characteristics of the AMX membrane in a 0.02 M NaCl solution; *i*_tot_ is the total current density, $i_{Cl^{-}}^{AEM}$ and $i_{OH^{-}}^{AEM}$ are the current densities of the Cl^−^ and OH^−^ ions through the membrane, respectively. The dashed lines show the tangents to the CVC used to determine the experimental limiting current density.

**Figure 2 ijms-20-03593-f002:**
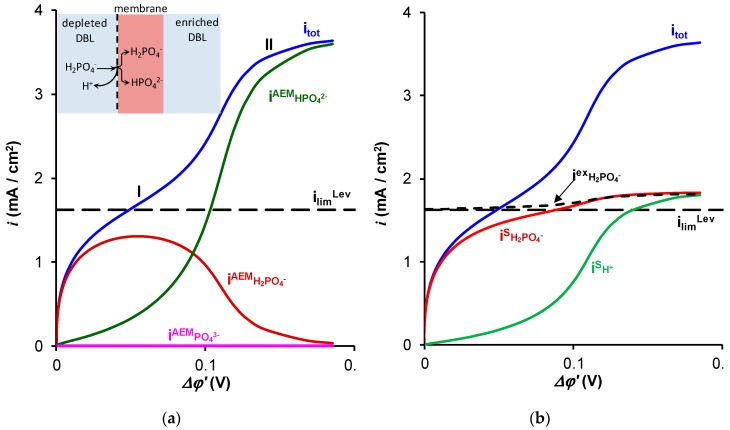
Theoretical total current density (i_tot_) and partial currents of H_2_PO_4_^–^ ($i_{H_{2}PO_{4}{}^{-}}^{AEM}$) and HPO_4_^2–^ ($i_{HPO_{4}{}^{2-}}^{AEM}$) ions in an AMX membrane (**a**) as well as the partial currents of H_2_PO_4_^–^ ($i_{H_{2}PO_{4}{}^{-}}^{s}$) and H^+^ ($i_{H^{+}}^{s}$) ions in the depleted solution at the membrane surface (**b**) as functions of the corrected potential drop. Solid lines were calculated using the model [31,32]. Dashed lines show the limiting current $i_{\lim}^{Lev}$ calculated using the Leveque equation, Equation (6), and the exaltation current, $i_{H_{2}PO_{4}{}^{-}}^{ex}$, calculated using Equation (1). “I” and “II” show the first and second inclined plateaus, respectively.

**Figure 3 ijms-20-03593-f003:**
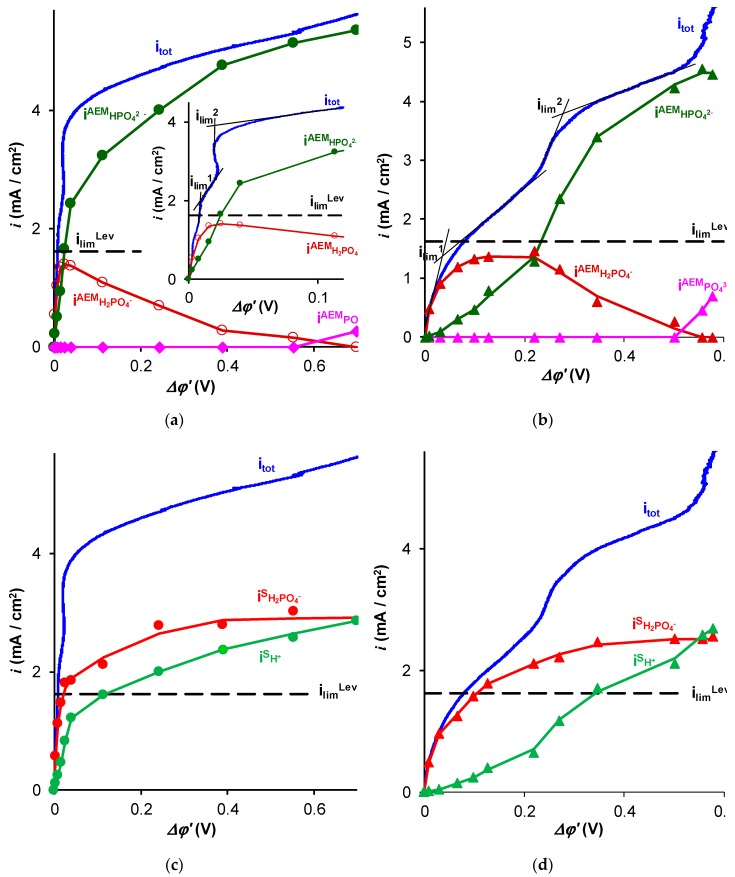
Experimental total current density (*i*_tot_) and partial currents of H_2_PO_4_^–^ ($i_{H_{2}PO_{4}{}^{-}}^{AEM}$), HPO_4_^2–^ ($i_{HPO_{4}{}^{2-}}^{AEM}$), and PO_4_^3–^ ($i_{PO_{4}{}^{3-}}^{AEM}$) ions in AMX (**a**) and AEM Fuji Type X (**b**) membranes, as well as partial currents of H_2_PO_4_^–^ ($i_{H_{2}PO_{4}{}^{-}}^{s}$) and H^+^ ($i_{H^{+}}^{s}$) in depleted solution near AMX (**c**) and AEM Fuji Type X (**d**) membranes as functions of the corrected potential drop. The curves connecting the markers are the fitting lines. The data were obtained in a 0.02 M NaH_2_PO_4_ solution. The dashed lines show the limiting current $i_{\lim}^{Lev}$ calculated using Equation (6).

**Figure 4 ijms-20-03593-f004:**
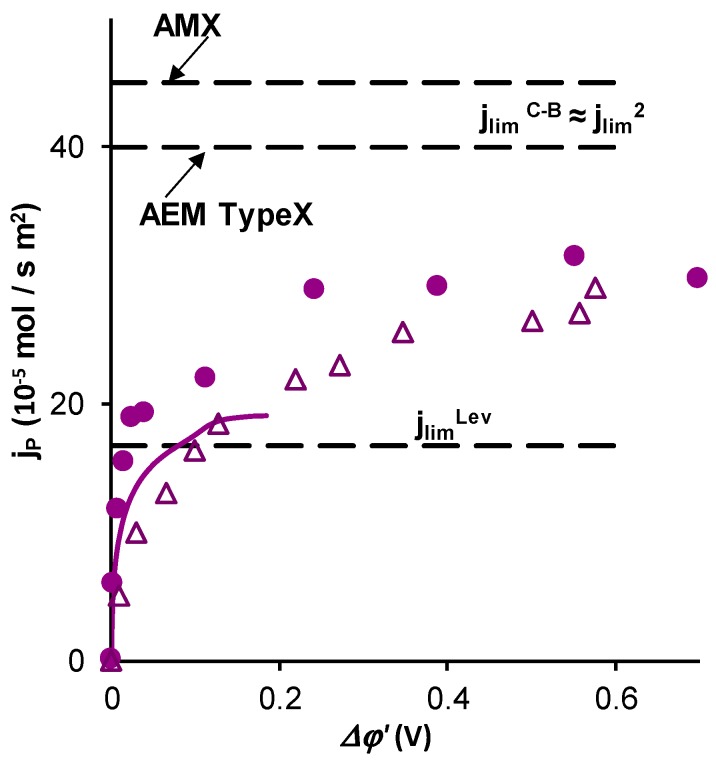
Experimental (markers) and calculated using the model [31,32] (solid line) total phosphorus flux density through the AMX (circles) and AEM Type-X (triangles) membrane vs. the reduced potential drop. Here *j*_lim_^Lev^
*= i*_lim_^Lev/^*F*, where *i*_lim_^Lev^ is calculated using Equation (6); *j*_lim_^C-B^ = *i*_lim_^C-B/^*F*, where *i*_lim_^C-B^ is found from the CVC curves by the Cowan–Brown method [43].

**Figure 5 ijms-20-03593-f005:**
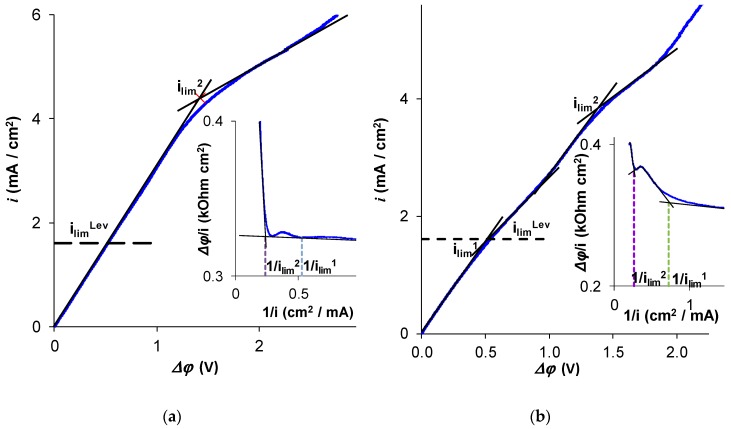
Current–voltage characteristics presented in the usual coordinates and the coordinates proposed by Cowan and Brown [43] (the insertion) for an AMX (**a**) and AEM Type X (**b**) membranes. *i*_lim_^1^ and *i*_lim_^2^ were the first and the second limiting currents determined by the tangent intersection method. The dashed lines in the insertions show the positions of the 1/*i*_lim_^1^ and 1/*i*_lim_^2^ points on the Cowan–Brown curves. The experiments were carried out in a 0.02 M NaH_2_PO_4_ solution.

**Figure 6 ijms-20-03593-f006:**
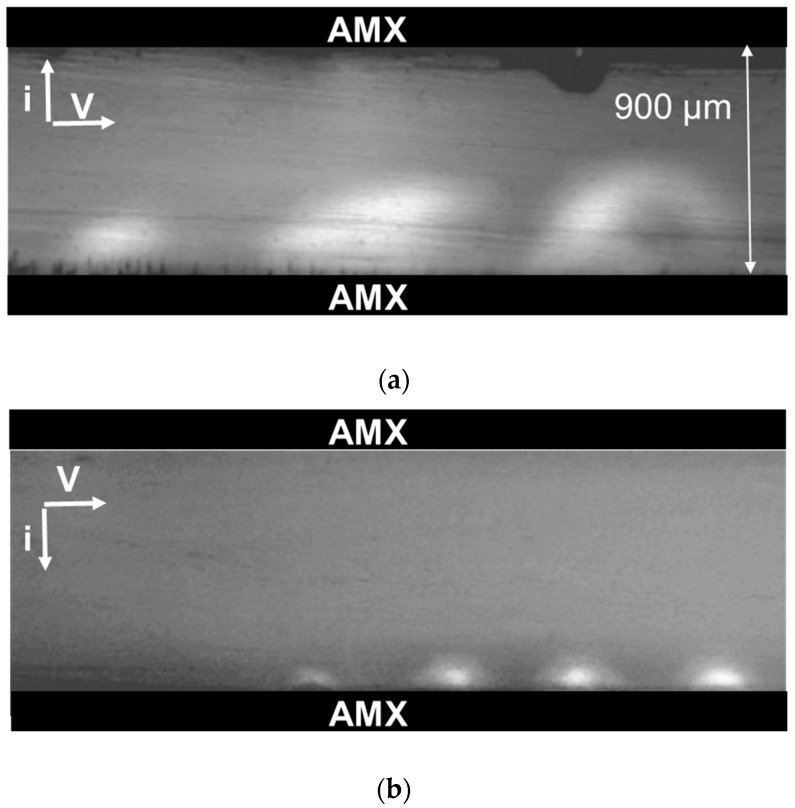
Visualization of the vortex structures at the AMX membrane surface in the desalination channel of the electrodialysis cell, where a 0.01 M NaCl solution (**a**) or a 0.01 M NaH_2_PO_4_ (**b**) was pumped. The intermembrane distance was *h* = 0.9 mm, the channel length was *L* = 3 mm, the linear flow velocity of the solution was *V* = 1 mm/s; *i/i*_lim_^Lev^ = 6.

**Figure 7 ijms-20-03593-f007:**
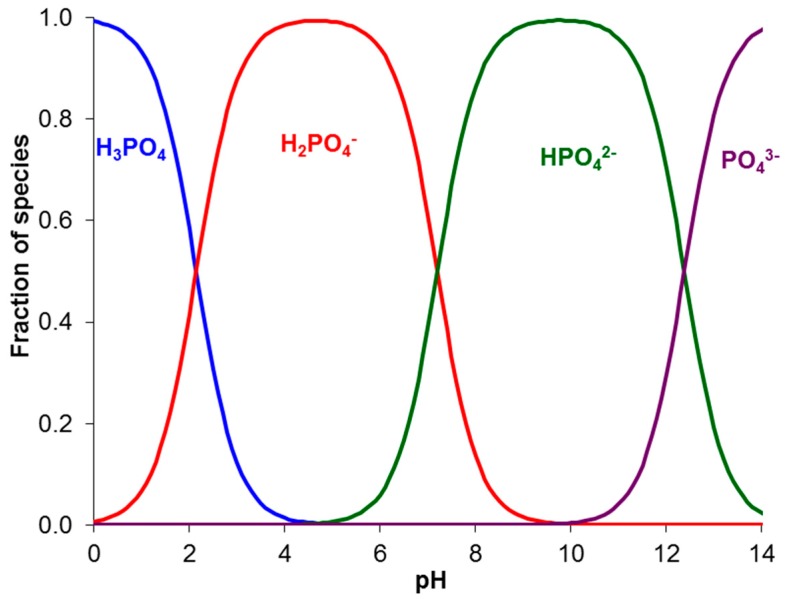
Distribution of the phosphoric acid species (in mole fractions) in a solution as a function of pH.

**Figure 8 ijms-20-03593-f008:**
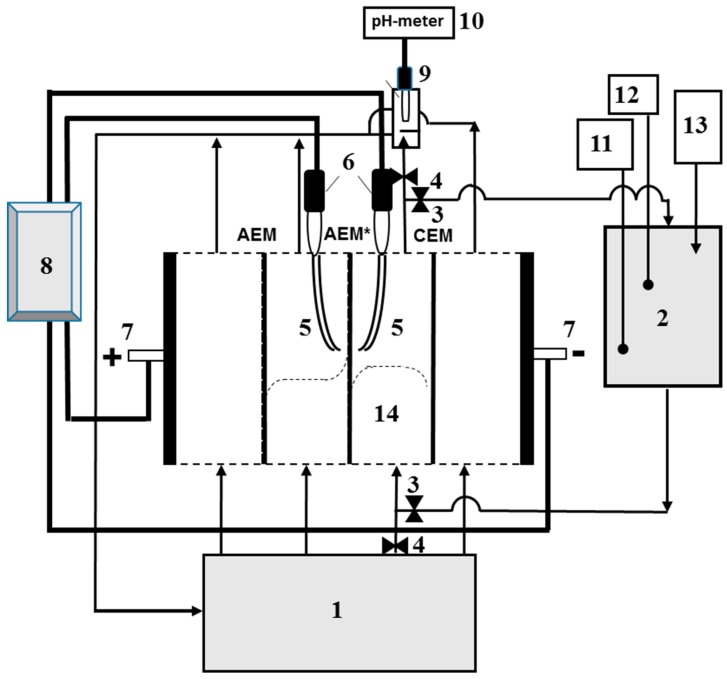
Schematic design of the setup used for determining the mass transfer and electrochemical characteristics of the cation-exchange membrane (CEM) and anion-exchange membrane (AEM) forming the desalination compartment (14). The setup included: an intermediate feed tank (1); an additional tank (2) for maintaining a constant pH; valves (3, 4); Luggin capillaries (5) connected with measuring Ag/AgCl electrodes (6); platinum polarizing electrodes (7); an electrochemical complex (an Autolab PGSTAT-100) (8); a flow cell (9) with an immersed combined electrode for pH measurement; a pH meter (10) connected to a computer; a combined electrode for pH measurement (11) connected to a pH meter; a conductivity cell (12) connected to a conductometer; a device (13) for maintaining a constant pH in the solution circulating through tank (2); AEM^*^ is the anion-exchange membranes under study; CEM and AEM are the auxiliary membranes. The dotted lines schematically show the counterion concentration profiles in the cell compartments.

**Table 1 ijms-20-03593-t001:** Several characteristics of the electrolyte solutions and membrane systems under study. The data are related to the temperature, 25 °C.

Electrolyte, pH	Diffusion Coefficients at Infinite Dilution, *D_i_*, 10^−5^ cm^2^s^−1^			Transport Numbers at Infinite Dilution		Theoretical Limiting Current Density in 0.02 M Solution ^*^, mA cm^−2^	Diffusion Layer Thickness ^*^, μm
	Cation	Anion	Electrolyte	Cation	Anion		
NaCl pH = 5.70 ± 0.05	1.334 [61,74]	2.032 [61]	1.61 [75]	0.396	0.604	3.07	256
NaH_2_PO_4_ pH = 4.60 ± 0.05		0.959 [61]	1.11 [74]	0.581	0.419	1.62	225

* Calculated using the Leveque equation (Equation (6)) as described in the Material and Methods section.

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
