# Peer review of "Partial Fluxes of Phosphoric Acid Anions through Anion-Exchange Membranes in the Course of NaH2PO4 Solution Electrodialysis"

_ijms, 2019, doi:10.3390/ijms20143593_

Round 1

Reviewer 1 Report

The article under review  “Partial fluxes of phosphoric acid anions through anion-exchange membranes in the course of NaH2PO4 solution electrodialysis “ by O. Rybalkina et al. has very high level regarding the presented results and their deep analysis but the text requires minor revision. There are some mistakes and incorrect sentences. English should be improved according to the following comments:

1.      P.1, line 13 (for the first time) and many times in the text of the article. It is not correct to write “…phosphorus transport through anion-exchange membranes…”.  The authors should use, for example, “…phosphorus-bearing species…” or “…all anions of phosphoric acid…”

2.      P.1, lines 27. It is necessary to write “…transforms…” instead of “…transform…”

3.       P. 1, lines 29-30. The sentence beginning with “Thus, the phosphorus ...” has no sense.

4.      P.1, line 40. “ion-exchange membrane” should be instead of “ion exchange membrane”

5.      P.2, line 48. It is necessary to change places of the words “electrodialysis” and “electrophoresis”.

6.      P.2, lines 49-51. It is not clear how ” …to extract these substances….from the products of their purification, separation and concentration.”

7.      P.2, line 71. It is necessary to write “…which are formed…” instead of “…which are forms…”

8.      P.3, line 107. Please, add word “acid” after word “phosphoric…”.

9.      P.4, line 153. “…are related to…”should be instead of “…relate to…”

10.   P.6, line 202. Different abbreviations (CEM, AEM) in the text and in the Fig.2 (KOM, AOM).

11.   P.8, line 268. Insert the reference to Eq.(8), not (5).

12.   P.8, line 272. “…it follows…” should be instead of “… it is follows…”.

13.   P.8, line 277. “… are determined,…” should be instead of “… is determined,…”.

14.   P.9, line 310. “…this layer contains..” should be instead of “… this layer contain…”.

15.   P.11, line 340. “…which width…” should be instead of “… whose width…”.

16.   P.11, line 347.”…are used to…” should be instead of “…used to…”.

17.   P.12, line 410. “…is limited by…” should be instead of “…is limiting by…”.

18.   P.14, line 437. “…is observed…” should be instead of “… are observed…”.

19.   P.15, line 449. “As Figure 5 shows…” should be instead of “ As Figure, 5 show…”.

20.   P.15, line 466. “The comparison…shows that…” should be instead of “The comparison…show that…”.

21.   P.16, line 476. “… is found from…” should be instead of “…is found form…”

22.   P.16, line 477. “…presented in usual coordinates…” should be instead of “…present in usual coordinates…”

23.   P.17, line 498. It is not correct to write “…flux of phosphorus atoms through the membrane”.

24.   P.17, line 505. Micrometer is not reflected in the text.

25.   P.18, lines 545,547,553,568,569. Problems with reflecting of some symbols.

26.   P.18, line 557. “…determines the rate…” should be instead of “…determine the rate..”

27.   P.19, line 579. “…return to…and participate in…” should be instead of “ …returns to…and participates in…”

28.   P.19, line 608. “…presented in the feed solution.” Should be instead of “…present in the feed solution.”

29.   P.20, line 642. “electrodialysis” should be instead of “selectrodialysis”

Author Response

Comments and Suggestions for Authors

The article under review  “Partial fluxes of phosphoric acid anions through anion-exchange membranes in the course of NaH2POsolution electrodialysis “ by O. Rybalkina et al. has very high level regarding the presented results and their deep analysis but the text requires minor revision. There are some mistakes and incorrect sentences. English should be improved according to the following comments:

1.      P.1, line 13 (for the first time) and many times in the text of the article. It is not correct to write “…phosphorus transport through anion-exchange membranes…”.  The authors should use, for example, “…phosphorus-bearing species…” or “…all anions of phosphoric acid…”

Dear Reviewer,

Thank you for your high evaluation of our paper. All the co-authors are sincerely grateful to you for the comments. Especially, I appreciate your remark on the erroneous number of equation. It is really hard to discern. And we are sorry to make so much misprint and errors.

As for this comment, thank you again. You are right: the phosphorus as a single species does not pass across the membrane, it is transported by the anions of phosphoric acid. To emphasize this fact, we revise the beginning of the Abstract:   “The phosphorus is transported through an anion-exchange membrane (AEM) by anions of phosphoric acid. However, which phosphoric acid anions carry the phosphorus in the membrane and boundary solution, that is, the mechanism of phosphorus transport, is not clear.

And hereinafter we use the terms you suggest: “…phosphorus-bearing species…” and “…all anions of phosphoric acid…” However, we keep also the term “phosphorus flux across the membrane” defining it as the sum of the fluxes of all phosphorus-bearing species.

2.      P.1, lines 27. It is necessary to write “…transforms…” instead of “…transform…”

Thanks, it is corrected.

3.      P. 1, lines 29-30. The sentence beginning with “Thus, the phosphorus ...” has no sense.

The sentence is corrected: “Thus, the phosphorus total flux, jP (equal to the sum of the fluxes of all phosphorus-bearing species) is limited…

4.      P.1, line 40. “ion-exchange membrane” should be instead of “ion exchange membrane”

Thanks, it is corrected.

5.      P.2, line 48. It is necessary to change places of the words “electrodialysis” and “electrophoresis”.

Thanks, it is corrected.

            6.      P.2, lines 49-51. It is not clear how ” …to extract these substances….from the products of their purification, separation and concentration.”

The sentence is corrected: "Electrophoresis and electrodialysis with ion-exchange membranes (IEMs) are used increasingly to extract these substances from wastewater [1-4], products of biomass processing [5], as well as the liquid wastes of food industry [6-11]. "

            7.      P.2, line 71. It is necessary to write “…which are formed…” instead of “…which are forms…”

Thanks, it is corrected.

              8.      P.3, line 107. Please, add word “acid” after word “phosphoric…”.

Thanks, it is corrected.

               9.      P.4, line 153. “…are related to…”should be instead of “…relate to…”

Thanks, it is corrected.

               10.   P.6, line 202. Different abbreviations (CEM, AEM) in the text and in the Fig.2 (KOM, AOM).

Thanks, it is corrected.

                11.   P.8, line 268. Insert the reference to Eq.(8), not (5).

Thanks, it is corrected. I appreciate this remark, as it is not easy to find an error in the equation number!

            12.   P.8, line 272. “…it follows…” should be instead of “… it is follows…”.

Thanks, it is corrected.

            13.   P.8, line 277. “… are determined,…” should be instead of “… is determined,…”.

Thanks, it is corrected.

            14.   P.9, line 310. “…this layer contains..” should be instead of “… this layer contain…”.

Thanks, it is corrected.

            15.   P.11, line 340. “…which width…” should be instead of “… whose width…”.

Thanks, it is corrected.

            16.   P.11, line 347.”…are used to…” should be instead of “…used to…”.

Thanks, it is corrected.

            17.   P.12, line 410. “…is limited by…” should be instead of “…is limiting by…”.

Thanks, it is corrected.

              18.   P.14, line 437. “…is observed…” should be instead of “… are observed…”.

Thanks, it is corrected.

            19.   P.15, line 449. “As Figure 5 shows…” should be instead of “ As Figure, 5 show…”.

Thanks, it is corrected.

            20.   P.15, line 466. “The comparison…shows that…” should be instead of “The comparison…show that…”.

Thanks, it is corrected.

            21.   P.16, line 476. “… is found from…” should be instead of “…is found form…”

Thanks, it is corrected.

            22.   P.16, line 477. “…presented in usual coordinates…” should be instead of “…present in usual coordinates…”

Thanks, it is corrected.

            23.   P.17, line 498. It is not correct to write “…flux of phosphorus atoms through the membrane”.

Thanks, it is corrected.

            24.   P.17, line 505. Micrometer is not reflected in the text.

Thanks, it is corrected.

            25.   P.18, lines 545,547,553,568,569. Problems with reflecting of some symbols.

Thanks, it is corrected.

             26.   P.18, line 557. “…determines the rate…” should be instead of “…determine the rate..”

Thanks, it is corrected.

            27.   P.19, line 579. “…return to…and participate in…” should be instead of “ …returns to…and participates in…”

Thanks, it is corrected.

            28.   P.19, line 608. “…presented in the feed solution.” Should be instead of “…present in the feed solution.”

Thanks, it is corrected.

            29.   P.20, line 642. “electrodialysis” should be instead of “selectrodialysis”

“selectrodialysis” is in the original title of the paper

With best regards, on behalf of all co-authors,

Victor Nikonenko

Reviewer 2 Report

In this manuscript, mainly the partial currents of different phosphoric species through AEMs were reported, along with current distribution between H2PO4-and H+in depleted layer of the solution. The experimental data was compared with the theoretical calculation. Very interesting current-voltage characteristics were obtained and well discussed. Visualization test of the impact of electroconvection is a merit to further validate the previous findings. The contents presented in the manuscript discuss important aspects of ion transport through IEMs coupled with chemical reaction inside the membrane in electrodialysis process, being high interest in electrochemical separation field. Overall, the manuscript is very systematic and well written, and I recommend it for publication in its current form after some minor spell checks, e.g., Line 174, should be "Figure 2"; In Figure 2, Define "AOM, KOM" in the graph, or should be "AEM, CEM" instead; Line 505, check the unit, etc.

Author Response

Comments and Suggestions for Authors In this manuscript, mainly the partial currents of different phosphoric species through AEMs were reported, along with current distribution between H2PO4-and H+in depleted layer of the solution. The experimental data was compared with the theoretical calculation. Very interesting current-voltage characteristics were obtained and well discussed. Visualization test of the impact of electroconvection is a merit to further validate the previous findings. The contents presented in the manuscript discuss important aspects of ion transport through IEMs coupled with chemical reaction inside the membrane in electrodialysis process, being high interest in electrochemical separation field. Overall, the manuscript is very systematic and well written, and I recommend it for publication in its current form after some minor spell checks, e.g., Line 174, should be "Figure 2"; In Figure 2, Define "AOM, KOM" in the graph, or should be "AEM, CEM" instead; Line 505, check the unit, etc. Dear Reviewer, Thank you for your high evaluation of our paper and the errors found. Your suggestions are greatly appreciated. The errors are corrected. With best regards, on behalf of all co-authors, Victor Nikonenko

Reviewer 3 Report

I read the manuscript with interest, since I have also observed similar pH-dependent effects in other electromembrane processes, such as in Donnan dialysis with anions formed by the water dissociation of polyprotic acids. The group of Prof. Nikonenko is well known for its long term important contribution to the field of ion-exchange membrane-based processes, such as electrodialysis and this study is in line with this adding additional understanding on the phenomena associated with the transport of phosphates across two anion-exchange membranes with distinct properties. The results are both experimentally obtained and theoretically interpreted. An important practical outcome of the study is the indication of reduced current efficiency for phosphorous recovery at higher current densities because of the increasing contribution of H+ in the total current in the system. Overall, the manuscript is clear and the illustrative material is well organized and presented. The conclusions withdrawn are supported by the results and have both theoretical and practical significance. 

I have some comments / suggestions for improving the text as follows:

Line 5-7: It is not necessary to have the superscript 1, since all authors have the same affiliation

Line 7: Change to: is easily detectable

Line 71: Change to:   which are formed

Line 114 Change to: to (CVC) curves.

Line 181 : Change to: before returning

Figure 2. Change the Russian abbreviations AOM and KOM to AEM and CEM

Line 203: I think it will be more correct to add counter-ion to concentration profiles since the co-ions will be repelled in the diffusion boundary layers…

Line 218: Change to:  if the solution pH is…

Line 247: Change to: in all cases presented in the paper

Line 272: Change to:  it follows

Line 287: Change to superscripts

Line 293: Change to phosphorous-containing compounds (or species)

Line 295: Change to: remaining compounds (or species)

Line 299: Change to: hence the total concentration

Line 312: Change to: gets higher

Line 410: Change this current is limited by diffusion

Line 412: Change to: is much lower than…

Line 505: Correct the unit to micrometers!

Line 545 Add the nominator!

Line 547: Add the symbolic representation for potential drop between,   ,…

Lines 568 and 569. Add the missing symbols!

Author Response

Comments and Suggestions for Authors

I read the manuscript with interest, since I have also observed similar pH-dependent effects in other electromembrane processes, such as in Donnan dialysis with anions formed by the water dissociation of polyprotic acids. The group of Prof. Nikonenko is well known for its long term important contribution to the field of ion-exchange membrane-based processes, such as electrodialysis and this study is in line with this adding additional understanding on the phenomena associated with the transport of phosphates across two anion-exchange membranes with distinct properties. The results are both experimentally obtained and theoretically interpreted. An important practical outcome of the study is the indication of reduced current efficiency for phosphorous recovery at higher current densities because of the increasing contribution of H+ in the total current in the system. Overall, the manuscript is clear and the illustrative material is well organized and presented. The conclusions withdrawn are supported by the results and have both theoretical and practical significance.

I have some comments / suggestions for improving the text as follows:

Dear Reviewer,

Thank you for your high evaluation of our paper and the errors found. Your suggestions are greatly appreciated. The errors are corrected.

Line 5-7: It is not necessary to have the superscript 1, since all authors have the same affiliation

Line 7: Change to: is easily detectable

Thanks, it is corrected.

Line 71: Change to:   which are formed

Thanks, it is corrected.

Line 114 Change to: to (CVC) curves.

Thanks, it is corrected.

Line 181 : Change to: before returning

Thanks, it is corrected.

Figure 2. Change the Russian abbreviations AOM and KOM to AEM and CEM

Thanks, it is corrected.

Line 203: I think it will be more correct to add counter-ion to concentration profiles since the co-ions will be repelled in the diffusion boundary layers…

Thanks, it is corrected.

Line 218: Change to:  if the solution pH is…

Thanks, it is corrected.

Line 247: Change to: in all cases presented in the paper

Thanks, it is corrected.

Line 272: Change to:  it follows

Thanks, it is corrected.

Line 287: Change to superscripts

Thanks, it is corrected.

Line 293: Change to phosphorous-containing compounds (or species)

Thanks, it is corrected.

Line 295: Change to: remaining compounds (or species)

Thanks, it is corrected.

Line 299: Change to: hence the total concentration

Thanks, it is corrected.

Line 312: Change to: gets higher

Thanks, it is corrected.

Line 410: Change this current is limited by diffusion

Thanks, it is corrected.

Line 412: Change to: is much lower than…

Thanks, it is corrected.

Line 505: Correct the unit to micrometers!

Thanks, it is corrected.

Line 545 Add the nominator!

Thanks, it is corrected.

Line 547: Add the symbolic representation for potential drop between,   ,…

Thanks, it is corrected.

Lines 568 and 569. Add the missing symbols!

Thanks, it is corrected.

With best regards, on behalf of all co-authors,

Victor Nikonenko